# Efficacy of LaAg Vaccine Associated with Saponin Against *Leishmania amazonensis* Infection

**DOI:** 10.3390/vaccines13020129

**Published:** 2025-01-27

**Authors:** Mirian França de Mello, Patrícia de Almeida Machado, Pollyanna Stephanie Gomes, Gabriel Oliveira-Silva, Monique Pacheco Duarte Carneiro, Tadeu Diniz Ramos, Juliana Elena Silveira Pratti, Raquel Peralva, Luan Firmino-Cruz, Alda Maria Da-Cruz, Luciana Covre, Daniel Claúdio Oliveira Gomes, Bartira Rossi-Bergmann, Eduardo Fonseca Pinto, Alessandra Marcia da Fonseca-Martins, Herbert Leonel de Matos Guedes

**Affiliations:** 1Laboratório de Imunobiotecnologia, Instituto de Microbiologia Paulo Góes, Universidade Federal do Rio de Janeiro, Rio de Janeiro 21941-902, RJ, Brazilpatricia_machado@id.uff.br (P.d.A.M.); pollyanna.gomes@ioc.fiocruz.br (P.S.G.); monique@biof.ufrj.br (M.P.D.C.); tdramos@micro.ufrj.br (T.D.R.); juliana_pratti@hotmail.com (J.E.S.P.); luanfcruz@micro.ufrj.br (L.F.-C.); alemartins@micro.ufrj.br (A.M.d.F.-M.); 2Laboratório de Imunologia Clínica, Instituto Oswaldo Cruz, Fundação Oswaldo Cruz, Rio de Janeiro 21040-360, RJ, Brazil; 3Laboratório Interdisciplinar de Pesquisas Médicas, Instituto Oswaldo Cruz, Fundação Oswaldo Cruz, Rio de Janeiro 21040-360, RJ, Brazilalda@ioc.fiocruz.br (A.M.D.-C.);; 4Laboratório de Imunobiologia, Núcleo de Doenças Infecciosas/Núcleo de Biotecnologia, Universidade Federal do Espírito Santo, Vitória 29047-105, ES, Brazildgomes@ndi.ufes.br (D.C.O.G.); 5Laboratório de Imunofarmacologia, Instituto de Biofísica Carlos Chagas Filho, Universidade Federal do Rio de Janeiro, Rio de Janeiro 21941-902, RJ, Brazil; bartira@biof.ufrj.br

**Keywords:** vaccine, *Leishmania*, saponin

## Abstract

Background/Objectives: The total lysate of *Leishmania amazonensis* (LaAg) is one of the most extensively studied vaccine formulations against leishmaniasis. Despite demonstrating safety and immunogenicity when administered intramuscularly, LaAg has failed to show efficacy in clinical trials and, in some cases, has even been associated with an enhanced susceptibility to infection. Adjuvants, which are molecules or compounds added to antigens to enhance the immunogenicity or modulate the immune response, are frequently employed in vaccine studies. This study aimed to evaluate different adjuvants to improve the protective efficacy of LaAg in *L.amazonensis* infection using a BALB/c mouse model. Methods: BALB/c mice were immunized with LaAg in combination with various adjuvants. The delayed-type hypersensitivity (DTH) test was assessed by measuring the infected paw and was used to evaluate the immunogenicity and to determine the most effective adjuvant. The immune response was analyzed through flow cytometry, focusing on cytokine production, immune cell recruitment and lesion size, alongside the control of parasite load at the infection site. The expression levels of iNOS and TGF-β were quantified using RT-qPCR, while IgG1, IgG2a and IgE antibody levels were determined via ELISA. Results: Among the adjuvants tested, only saponin (SAP) elicited a significant DTH response following LaAg challenge. SAP enhanced the immunogenicity of LaAg, as evidenced by increased IFN-γ-producing CD4^+^ and CD8^+^ T cells in the draining lymph nodes at 18 h post-challenge. Additionally, SAP facilitated the recruitment of lymphocytes, macrophages, neutrophils and eosinophils to the infection site. Conclusions: The LaAg + SAP combination conferred partial protection, as demonstrated by a reduction in lesion size and the partial control of parasite load. In conclusion, the addition of SAP as an adjuvant to LaAg effectively modulates the immune response, enhancing the vaccine’s protective efficacy. These findings provide valuable insights into the development of improved vaccines against *L.amazonensis* infection.

## 1. Introduction

Leishmaniasis is one of the major neglected tropical diseases. It is caused by at least 20 parasitic species of the genus *Leishmania* which are transmitted between mammalian hosts by female sandflies. Two main clinical forms can be found as follows: cutaneous leishmaniasis (CL) and visceral leishmaniasis (VL) [1]. Today, more than 1 billion people live in areas where leishmaniasis is endemic and are at risk of infection. An estimated 50,000 to 90,000 new cases of VL and 600,000 to 1 million new cases of CL occur annually [2]. Treatment of leishmaniasis relies on a few chemotherapeutic agents, such as pentavalent antimonials, amphotericin B, miltefosine, paromomycin and pentamidine, most of which are parenteral in use and induce severe adverse effects. Furthermore, rates of treatment failure are high and have been linked to drug resistance [3]. While Ambisome^®^ (liposomal amphotericin B) has demonstrated unparalleled efficacy in recent years, its high cost per dose limits its widespread use, particularly in endemic regions [4]. Presently, there is no ideal vaccine or drug to eradicate leishmaniasis.

Currently, the vaccines against leishmaniasis that are being studied can be divided into first-, second- and third-generation vaccines. First-generation vaccines are composed of killed or attenuated microorganisms, or even uncharacterized mixtures of antigens. Second-generation vaccines can be composed of defined subunits. Third-generation vaccines are genetic vaccines using DNA or RNA platforms, composed of the gene of the disease-causing agent that encodes an immunogenic antigen [5]. Leishvacin^®^, a first-generation vaccine developed in Brazil for human use, is composed in the first moment of the promastigote lysates of five strains of *Leishmania* and then selects one strain of *L.amazonensis*. Although phase I clinical studies have attested to its safety [6] and phase II clinical studies have attested to its immunogenicity [7], the use of this vaccine was discontinued due to its lack of efficacy in phase III clinical studies carried out in Colombia based on the incidence of individuals that developed CL caused by *L.panamensis* (7.7% in the vaccinated group versus 6.8% in the placebo group) [8]. It was previously demonstrated that the intramuscular vaccination of BALB/c mice with a total lysate of *L.amazonensis* (LaAg) results in an increased susceptibility to CL [9]. In contrast, oral vaccination with LaAg protects BALB/c and C57BL/6 mice against CL [10]. However, the intrahepatic route has shown inefficiency in generating a robust immune response, making it a less favorable method for inducing protective immunity against CL in BALB/c mice [10]. Besides the vaccination route, vaccine effects can be improved from the association of antigens with adjuvants.

Adjuvants may increase the immunogenicity of antigens stimulating protective immunity based on antibody production and the induction of effector T cells, while also causing less toxicity or longer lasting immunological effects [11,12]. Various adjuvants have been researched for leishmaniasis vaccines, with some already used in existing vaccines. Freund’s adjuvant, although potent, is known for its toxicity and is not widely used in human vaccines. Aluminum hydroxide (Alum) is one of the most commonly used adjuvants due to its safety profile and effectiveness in inducing a TH2 response, although it may be less effective for intracellular pathogens like *Leishmania* [10,12]. Addavax™, a squalene-based oil-in-water emulsion similar to MF59, has shown promise in enhancing both antibody and T cell responses. Saponin-based adjuvants, like QS-21, have also been explored for their ability to stimulate a strong Th1 response, which is crucial for protection against *Leishmania* infection [13]. Montanide ISA, another oil-based adjuvant, has been used in various experimental vaccines due to its ability to enhance both cellular and humoral responses. Monophosphoryl Lipid A (MPLA), a detoxified derivative of lipopolysaccharide, has been used to boost the immune response by activating toll-like receptor 4 (TLR4) and promoting a Th1 response, which is particularly beneficial for vaccines against intracellular pathogens [11,13,14].

As demonstrated in several studies, for example, using *Leishmania* antigens, specifically *L.braziliensis* combined with saponin, is a vaccination strategy that has led to significant improvements in both homologous and heterologous challenge protection and has resulted in a reduced parasite load during the course of infection. Effective control has been achieved, associated with a robust cellular response compatible with effective parasite control. Thus, our objective was to evaluate whether the association of LaAg with different adjuvants could induce protection, and to characterize the protection mechanism of the most effective combination.

## 2. Materials and Methods

Animals: Female BALB/c WT mice, aged 8 to 10 weeks, were acquired from the Institute of Science and Technology in Biomodels (ICTB). All animals were used according to the “Basic Principles for Research Involving Animals”, approved by CEUA FIOCRUZ L027/2020.

Parasites: *Leishmania amazonensis* (MHOM/BR/75/Josefa) expressing GFP [8,9] were maintained at 26 °C in M199 medium (Cultilab, Campinas, SP, Brazil) containing 10% heat-inactivated fetal bovine serum (FBS, Cultilab, SP, Brazil), hemin (5 μg/mL, Sigma-Aldrich, St. Louis, MO, USA) and 2% sterile human urine. Amastigotes were frequently isolated from the lesions of BALB/c mice for maintaining the infectivity of the strain.

Preparation of *L.amazonensis* total lysate (LaAg): *Leishmania amazonensis* promastigote antigens (LaAg) were prepared as previously described [10]. Briefly, stationary-growth phase promastigotes were washed three times in phosphate-buffered saline (PBS) and subjected to three cycles of freezing and thawing. Lysis was evaluated by observing the absence of whole parasites under the microscope. The protein concentration was assessed using the Lowry assay (Bio-Rad, Hercules, CA, USA). LaAg was lyophilized, stored at −20 °C and reconstituted with PBS immediately prior to use.

Intramuscular vaccination (I.M.): BALB/c mice were vaccinated with LaAg alone (100 μg) or in combination with different adjuvants: CFA (Sigma Aldrich, St. Louis, MO, USA) (80 µL/dose), Saponin (Sigma Aldrich, St. Louis, MO, USA) (SAP), Riedel de Haen’s (100 μg/dose) or AddaVax™ (Invivogen, San Diego, CA, USA) (50 µL/dose) and PBS (100 µL—control group). Mice received two doses of the vaccine by the intramuscular route (quadriceps muscle), with an interval of 7 days between them. After determining the association that generated the greatest cutaneous hypersensitivity response (using a caliper), we decided to investigate only the LaAg + SAP association. Thereafter, BALB/c mice (*n* = 5 animals per group) were immunized by the intramuscular route with two doses in 100 μL LaAg (100 μg in PBS), SAP (100 μg in PBS) or LaAg + SAP (100 μg of LaAg + 100 μg of SAP in PBS), or 100 μL of PBS (control group), with an interval of 7 days between the two doses. The same protocol of vaccination and infection was followed for experiments of immunogenicity, immunomodulation and efficacy. In the immunogenicity experiments, mice were euthanized 3 days after the second vaccine dose; in immunomodulation experiments, euthanasia occurred 18 h after infection/challenge (peak of hypersensitivity) and in experiments of vaccine efficacy, the development of the lesion was monitored until day 50, when the animals were euthanized.

Challenge with LaAg or *L.amazonensis*: To conduct immunomodulation and vaccine efficacy experiments, seven days after the second vaccine dose, the animals were challenged with either LaAg (10 μg) or stationary-phase *L.amazonensis* promastigotes (2 × 10^6^) in 10 μL of PBS on the right footpad, using a 50 μL microsyringe (Hamilton, CA, USA) and Ultrafine needle (Becton, Dickinson and Company, Franklin Lakes, NJ, USA).

Evaluation of vaccine efficacy: The vaccine efficacy was evaluated by lesion progression and parasitic load analysis. After immunization and challenge, lesion sizes were measured once a week with a thickness gauge (Mitutoyo 7301) and expressed as the difference between the thicknesses of infected and contralateral non-infected footpads. The parasite load was determined at the end of the experiment (day 50), when the infected footpad was removed and individually homogenized in 1 mL of PBS using a tissue grinder. Tissue debris was eliminated through a 5 min gravity sedimentation process. Homogenates were submitted to limited dilution assay (LDA), in which the dilutions of the supernatants were plated in triplicates on a 96-well black plate. Analysis of the parasite load was also performed on infected footpads by fluorometry [15] and the fluorescence intensity was read in a BIO TEK plate fluorimeter at excitation and emission wavelengths of 435 nm and 538 nm, respectively. The parasite load was considered directly proportional to the fluorescence of the macerate [10].

Vascular permeability: BALB/c mice previously immunized with two doses of LaAg + SAP or the PBS control received the Evans blue dye (20 mg/kg) through the orbital plexus one hour before infection. The mice were euthanized 18 h after the infection and the footpads were removed. The intact footpads were weighed, incubated in formamide (4 mL/g of footpad) for 24 h at room temperature and then the spectrophotometer was read at 650 nm. The data obtained reflected the subtraction of the reading from the infected footpads by the uninfected footpads.

Histology: BALB/c mice were euthanized 18 h after infection and the footpads were sectioned uniformly to obtain the histological sections. The infected footpads were fixed in 10% paraformaldehyde and decalcified in 14% EDTA. Sections of 4 μm were made, followed by staining with hematoxylin and eosin and finally cell counting.

Real time PCR (RT-qPCR): Relative quantifications of the mRNA transcripts of TGF-β and iNOS were made using RT-qPCR. The footpads were incubated in 4 M guanidine solution and macerated in a tissue homogenizer (Ika Company, NC, USA). To the homogenized material was added the 0.1 time (2 M) volume of sodium acetate pH4, an equal volume of DEPC phenol and 0.3 times the volume of chloroform. The samples were incubated on ice for 20 min and then centrifuged at 2000 G/4 °C for 30 min. The aqueous phase, containing the RNA, was transferred to a new tube, where 1.1 times the volume of isopropanol was added. The samples were incubated for 16 h at −20 °C and then centrifuged again at 2000× *g*/4 °C for 30 min. The supernatant was discarded, and 70% ethanol was added to the pellet. The samples were again centrifuged. To the obtained pellet, water treated with DEPC was added. The total RNA samples obtained were dosed in NanoDrop (Thermo Scientific, CA, USA) at 260 nm. For cDNA preparation, 1 μg of RNA was treated with DNAse (2 units, Invitrogen, Carlsbad, CA, USA) for 30 min at 37 °C, followed by inactivation by the addition of 1 μL of 25 mM EDTA and incubation at 65 °C for 10 min. The SuperScript™ II Reverse Transcriptase kit (Invitrogen) was used, containing 1 μL of 50 ng/μL random primer, 1 μL of dNTP mix (10 mM each nucleotide), 4 μL of enzyme buffer, 2 μL of DTT (10 mM) and 1 μL of reverse transcriptase (200 units/μL). The reaction was incubated at 42 °C for 50 min, followed by heat inactivation at 70 °C for 15 min. The cDNA obtained was stored at −80 °C. For each reaction, 1 μL of the cDNA obtained was used, along with 12 μL of SYBR^®^ Green PCR Master Mix (Applied Biosystems, Foster City, CA, USA), 2.5 mM of each primer (forward and reverse) and DEPC-treated water in a sufficient quantity for 25 μL of reaction. The primers used were designed with the program Primer Express^®^ version 3.0 (Applied Biosystems). Reactions were performed on the StepOne device (Applied Biosystems) with the StepOne™ software v2.1 program. The reaction conditions were as follows: 40 cycles at 95 °C/30 s and 60 °C/1 min. β-actin expression was used as the endogenous control and the control group PBS was used as the reference sample for the calculation of relative quantifications (RQ); the β-actin gene is widely recognized for its stability and conservation across different species and experimental conditions, making it a reliable reference gene for normalization in gene expression studies [16]. The relative quantification was performed using the ΔΔCt method, which assumes 100% efficiency of the qPCR assays [17].

Antibody detection: The blood collected from animals was centrifuged at 12,000× *g* in the centrifuge for 20 min. After centrifugation, the supernatant was removed, and the pellet formed was discarded. Serum was used for antibody analysis by ELISA. Then, 1 μg of LaAg was plated overnight in carbonate buffer. After washing and blocking, a serum dilution of 1:250 was employed for all samples. Then, the anti-IgG1, anti-IgG2a and anti-IgE antibodies were employed (1:5000 dilution). After 1 h, streptavidin was added, each well was covered with TMB and the reactions were stopped with sulfuric acid. The reading was performed at 450 nm.

Flow Cytometry: Popliteal lymph node (draining lymph node) cells were plated (2 × 10^6^ cells/well) in a 96-well round bottom plate. The lymph node cells were cultured for 4 h at 37 °C in the presence of PMA (20 ng/mL, Sigma Aldrich, St. Louis, MO, USA), Ionomycin (1 μg/mL, Sigma Aldrich) and brefeldin A (5 mg/mL, Sigma Aldrich). The cells were then washed and stained with anti-CD3-PERCP, anti-CD4-PECY7 and anti-CD8-FITC (Biolegend, San Diego, CA, USA); fixed with 1% paraformaldehyde and permeabilized with PBS and 0,02% saponin (permeabilization buffer). Intracellular cytokine staining was performed using anti-IFN-γ-APC and either anti-IL10-PE or anti-IL-4-PE (Biolegend) over 1 h. The cells were washed in permeabilization buffer and resuspended in FACS buffer (PBS with 5% fetal calf serum–Gibco) for flow cytometry readings (BD FACSCanto™ II–BD Biosciences, NJ, USA). The data were analyzed using the FlowJo X software v10.10.

Statistical analysis: The experiments were conducted two or three times, with the results from one representative experiment presented. The normal distribution was confirmed using the Shapiro–Wilk test, followed by parametric tests such One-way or Two-way ANOVA, with Bonferroni’s post-test. For bar graphs, statistics were determined by Student’s *t*-test. We used the GraphPad Prism v. 8 software, and differences were considered significant when *p* ≤ 0.05.

## 3. Results

### 3.1. Association of LaAg-Plus-Saponin-Induced Higher Delayed-Hypersensitivity (DTH)

Initially, in a tentative attempt to make intramuscular LaAg protective, we immunized mice with LaAg alone (100 μg/dose) or in combination with different adjuvants as follows: CFA (80 µL/dose), SAP (100 μg/dose) or AddaVax™ (50 µL/dose), and PBS was used as a control group, to determine which vaccine formulation induced DTH. BALB/c mice received two doses of the vaccine by the intramuscular route, with an interval of 7 days between them. Seven days after the second dose, the animals were challenged with 10 μg LaAg (Figure 1A) and/or 2 × 10^6^ *L.amazonensis* promastigotes in the stationary phase (Figure 1B) in the footpads, and the swelling was measured at 0, 18, 24 and 48 h post-challenge. The challenge with LaAg induced DTH only in the group vaccinated with LaAg + SAP when compared to the PBS control group. The increase in thickness was observed at times of 18, 24 and 48 h (Figure 1A). The challenge with *L.amazonensis* induced a DTH response in the LaAg + CFA, LaAg + SAP and LaAg + AddaVax™ groups when compared to the PBS control group, with a peak of DTH at 18 h post-challenge, and the LaAg + SAP group remained the most responsive (Figure 1B). When the animals were challenged with *L.amazonensis* and, seven days post-infection, challenged with LaAg in the contralateral footpad, the profile was modified indicating that infection modified the cellular response; however, mice immunized with LaAg + SAP continued to be the most responsive (Figure 1C). As the group of animals vaccinated with LaAg + SAP developed the higher DTH response, we decided to determine the possible protective mechanism of this association, as well as the vaccine efficacy of LaAg + SAP, which will be reported in the topics ahead.

### 3.2. Immunization with LaAg + SAP Induced A Th1 Profile

For the evaluation of immunogenicity induced by immunization with LaAg + SAP, mice were vaccinated with two doses of LaAg + SAP by the intramuscular route, with an interval of 7 days between them, and euthanized 3 days after the second vaccine dose. It was observed that the LaAg + SAP vaccine induced cellular lymphoproliferation, with an increase in CD4^+^ and CD8^+^ T cells, but did not induce a difference in the frequency of FoxP3+ Treg cells in the popliteal lymph nodes. Phenotypic analysis of T cells showed that the LaAg + SAP vaccine is capable of inducing IFN-γ-producing T cells, and both CD4^+^ T cells (Figure 2A,B) and CD8^+^ T cells (Figure 2C,D) produced more IFN-γ in the LaAg + SAP vaccine group than in the PBS control group. The immunogenicity data also showed an increase in IL-10-producing CD8^+^ T cells (Figure 2G,H), but not IL-10-producing TCD4^+^ (Figure 2E,F). In addition, there was a decrease in IL-4-producing TCD8^+^ (Figure 2K,L).

We also evaluated the production of the specific antibodies IgG1, IgG2a and IgE in blood plasma using ELISA. We observed that both LaAg alone and LaAg + SAP had a significant increase in IgG1 (Figure 3A) and IgG2a (Figure 3B), indicating a mixed Th1/Th2 response based on antibody production. When we measured IgE, we did not detect an alteration in IgE levels in blood plasma after vaccination (Figure 3C). As in the immunogenicity experiments, a direction towards the Th1 immune response was observed, so we decided to evaluate the DTH response after vaccination followed by challenge.

### 3.3. Mice Immunized with LaAg + SAP Induced Intense Cell Recruitment After Challenge with L.amazonensis

Subsequently, we characterized DTH induced by the LaAg + SAP vaccine after challenge with *L.amazonensis*. For this, the mice were vaccinated with two doses of the vaccine (LaAg + SAP) by the intramuscular route, with an interval of 7 days between them. Seven days after the last dose, the animals were challenged with *L.amazonensis* promastigotes. The footpads were photographed 18 h after infection (peak of DTH) and these data are represented in Figure 4A. It can be seen that vaccination with LaAg + SAP induced DTH in animals challenged with *L.amazonensis*, with an increase in footpad thickness in this group compared to the controls (PBS, LaAg and SAP), which is illustrated Figure 4A. These data corroborate the data previously presented in Figure 1.

To evaluate the vascular permeability of the tissue, comparing the PBS control and the LaAg + SAP vaccine groups, Evans blue dye (20 mg/kg) was administered intravenously (through the orbital plexus) 7 days after the last dose and 1 h before challenge. The mice were euthanized 18 h after the challenge, the footpads were removed, incubated in formamide for dye extraction and the samples were read in a spectrophotometer at 650 nm. The data obtained demonstrate a greater permeability of tissue in animals previously vaccinated with LaAg + SAP when compared to the PBS control group (Figure 4B). To assess the cellularity in the site of infection at the peak of DTH, we euthanized the mice 18 h after challenge and the footpads were sectioned uniformly to obtain the histological sections. There was an increase in the cellularity in the animals previously immunized with LaAg + SAP when compared to the control group PBS. The same did not occur with the LaAg or SAP control groups (Figure 5A). In the LaAg + SAP vaccine group, there was a significant increase in the following cells compared to the PBS group: neutrophils, eosinophils, macrophages and lymphocytes. The exception was mast cells, for which there were no changes between the PBS and LaAg + SAP (Figure 5F) groups. Using the same protocol, some footpads were used for the relative quantification of the mRNA transcripts of TGF-β and iNOS (Appendix A). Analysis of the transcripts revealed that the mRNA for TGF-β reduced, while the mRNA for iNOS increased more than 2-fold (Appendix A). To prove that DTH is dependent on T cells, we immunized RAG^-/-^ with LaAg SAP, and observed no DTH response in the absence of lymphocytes (Appendix A).

### 3.4. Mice Immunized with LaAg + SAP Induced Intense Th1 Response After Challenge with L.amazonensis

Eighteen hours post-infection, in the popliteal lymph nodes, which are the draining lymph nodes of the infection, we counted the cells and analyzed by flow cytometry to identify cell subtypes. The number of cells counted in Neubauer’s camera revealed that in the LaAg + SAP vaccine group there was an increase in cellularity when compared to the PBS control. The same phenomenon did not occur in any other group. FACS labeling revealed that this increase in cell numbers was accompanied by increased numbers of both CD4^+^ and CD8^+^ T cells, corroborating the immunogenicity data showed previously. Phenotypic analysis of popliteal lymph node cells revealed a significant increase in IFN-γ-producing CD4^+^ T cells (Figure 6A,B), and a more significant increase in IFN-γ-producing CD8^+^ T cells (Figure 6C,D). Although CD4^+^ T cells were the cells with the greatest expansion when compared to CD8^+^ T cells, it was only the CD8^+^ T cells that produced more IFN-γ, while in the LaAg + SAP vaccine group, the IFN-γ-producing CD4^+^ was 0.6% and the IFN-γ-producing CD8^+^ was 20% (Figure 6A–D). In addition to the increased participation of CD8^+^ T cells in the production of IFN-γ, these cells also produced other cytokines, such as IL-10 (Figure 6G,H) and IL-4 (Figure 6K,L) denoting a mixed response of the inflammatory profile after infection, predominantly directed by CD8^+^ T cells. When we evaluated the production of IL-4 and IL-10 by CD4+ T cells, we observed that there was production of IL-10 (Figure 6F) but not of IL-4 (Figure 6I,J), suggesting a more compromised profile with the Th1 response. As the combination of LaAg + SAP was shown to induce a cutaneous DTH response and to be immunogenic with significant IFN-γ production by CD4^+^ and CD8^+^ T cells, we investigated if this combination could be protective.

### 3.5. Immunization with LaAg + SAP Vaccine Induced A Partial Protection in BALB/c Mice Against L.amazonensis Infection

The mice were previously immunized according to the protocol described below, and one week after the second vaccine dose or PBS (control group), all groups were infected in the right footpad with 2 × 10^6^ promastigotes of *L.amazonensis* (GFP) at the stationary growth phase. The development of the lesion was monitored until day 50 by measuring the thickness of the footpad by pachymetry. At the end of the experiment, the animals were submitted to euthanasia and the parasite load in the lesion was evaluated. In Figure 7, we observed that immunization with LaAg + SAP in BALB/c mice induced partial protection, controlling both lesion growth (Figure 7A) and parasite load (Figure 7B), and LDA (Figure 7C) aligned with the results obtained through fluorescence when compared to the PBS control group.

## 4. Discussion

Currently, leishmaniasis is primarily controlled through chemotherapy, but its limitations—such as side effects, toxicity, resistance, high cost and low success rate—highlight the urgent need for an effective, long-lasting vaccine [18]. Although recovery from a primary infection provides some immunity to reinfection, no ideal vaccine has yet been developed, partly due to the absence of a suitable adjuvant [11,19].

Vaccination with LaAg has already been shown to be safe and immunogenic [7]; however, it failed with respect to efficacy by the subcutaneous, hepatic [10] and intramuscular [9] routes. The LaAg antigen was deleterious and had the capacity to induce cell anergy in vitro, whereas LbAg induced cell proliferation in cells from the draining lymph nodes infected with *L.amazonensis* [20]. When comparing LbAg versus LaAg by intramuscular route, LaAg increased lesion size and parasite load while LbAg induced non-effect against *L.amazonensis* infection on BALB/c [9].

Due to issues associated with the total antigen of *L.amazonensis* without adjuvant, some groups have opted to use the total antigen of *L.braziliensis* as an alternative to LaAg. Vaccination with LbAg in BALB/c mice did not increase their susceptibility to *L.amazonensis* infection [9]. One way to improve the effect of vaccines is to associate the antigen with adjuvants [12]. In almost all recently developed vaccines, adjuvants have been incorporated [21]. Specifically, for leishmaniasis, there are numerous benefits in the incorporation of adjuvants in vaccines, such as (1) enhancing immunogenicity and efficacy, (2) modulating T cell phenotypes toward Th1, (3) reducing the number of doses and (4) extending the induction of specific effector CD4^+^ and CD8^+^ T cells [11,22]. LbAg combined with SAP (LbSap) has been extensively studied against visceral leishmaniasis, proving to be a protective vaccine for dogs [23,24]. Based on that, the use of adjuvants can become LaAg-protective in vivo.

Studies on the protective response against *Leishmania* spp. infection have shown that local DTH is an indicator of protection and seems to be mediated by the cellular response [25,26,27,28]. In our study, we assessed the impact of combining LaAg with various adjuvants (CFA, SAP and AddaVax™) on DTH responses in BALB/c mice. Only the animals vaccinated with LaAg + SAP exhibited a DTH response after challenge with LaAg, compared to the PBS control group, and showed the highest DTH reaction among all groups evaluated after challenge with *L.amazonensis*. DTH reactions are correlated with Th1-producing IFN-γ and consequently with a protection against infection by *L.infantum* by previous sandfly bites in hamsters [29] and dogs [30].

A difference in the DTH profile was observed when comparing challenges with LaAg and live *Leishmania*. When live *Leishmania* was inoculated, a higher DTH response was observed compared to LaAg at 18 h post-infection, suggesting that the parasite modulates the immune response differently from its total antigen; however, using LaAg as challenge in the mice vaccinated with LaAg + SAP, it was observed that there was a higher DTH at 24 and 48 h post-challenge. This indicates that the parasite’s escape mechanisms are involved in modulating the cellular response. This observation prompted us to further investigate the underlying mechanisms of DTH and identify the key early events that may be crucial for the vaccine’s effectiveness.

We evaluated the immunogenicity and immunomodulation mechanisms of LaAg + SAP in BALB/c mice. We observed that the LaAg + SAP vaccine induced a cellular response, with an increase in CD4^+^ and TCD8^+^ T cells in the popliteal lymph nodes, showing that the protective mechanism was induced by LaAg + SAP. In addition, the LaAg + SAP vaccine was capable of inducing IFN-γ-producing T cells (CD4^+^ and CD8^+^) and decreasing IL-4-producing CD8^+^ T cells. This immune response profile favors the development of a Th1 response, which would be ideal for combating a *Leishmania* infection.

Our results corroborate what was seen in an experimental study that evaluated the potential protection of the LbAg + Sap + Sal vaccine in dogs at different time points (0, 90, and 885 days), which induced an increase in type I cytokines (TNF-α, IL-12, IFN-γ) and a reduction in type II cytokines (IL-4) and suppressor cytokines (TGF-β), even after challenge with *L.infantum* [31]. Thus, our proposed vaccine appears to be a promising tool for controlling infection by both in *L.infantum* [31] and the homologous challenge.

Regarding antibodies measured in blood plasma, LaAg + SAP induced an increase in IgG1 and IgG2a, suggesting a mixed Th1- and Th2-type immune response. The lack of a significant increase in IgG1 and IgG2a levels with the addition of SAP compared to the antigen alone may be explained by the high intrinsic immunogenicity of LaAg, which likely elicited a robust humoral response on its own, limiting the observable effect of SAP. Additionally, SAP may predominantly enhance cell-mediated immunity, aligning with the Th1-biased response observed, rather than significantly boosting antibody production. This suggests that SAP’s primary role might be in modulating the quality of the immune response rather than increasing antibody titers. The same was seen with formulations with *L.donovani* antigens and homologous challenge, which also induced the serum protection of IgG1 and IgG2a inducted of a mixed Th1/Th2 response after immunization in BALB/c mice [32]. Also seen were DTH and the increased production of NO and IFN-γ by spleen cells and the downregulation of IL-4, which demonstrates that an initial stimulation of a mixed Th1/Th2 response by vaccination instructs Th1 responses and resistance against a progressive infection [32]. Likewise, studies with vaccines in nanoformulations with *L.amazonensis* [33] or *L.braziliensis* [34] with heterologous challenge (*L.infantum*) in a hamster model induced strong antigenicity with increases in total IgG levels; however, there was no correlation with protection.

The production of IL-10 by CD8^+^ T cells could contribute to controlling the excessive inflammatory response that is associated to tissue damage [35]. The IL-10 from CD4^+^ and CD8^+^ cells in parasitic diseases has been described as delaying control while generating immunity due to treatments or vaccination [36]. According to our data, the increase in IL-10 from these same sources is correlated with the group that was partially protected. Thus, as demonstrated by Grenfell et al. (2010), BALB/c mice immunized with *L.amazonensis* and *L.braziliensis* antigens alone or in combination with SAP exhibited partial protection against the *L.infantum* and showed a significantly reduced production of IL-4 and IL-10 concomitantly [37]. A high IFNγ/IL-10 ratio is generally associated with resistance to *Leishmania* infections, as IFNγ is a marker of Th1 response, which is essential for controlling intracellular parasites. IL-10, on the other hand, is a regulatory cytokine that can suppress the Th1 response, thereby favoring parasite persistence that is essential for protection in Leishmanization [38]. Therefore, maintaining a high IFN-γ/IL-10 ratio may be crucial for protection against *L.amazonensis* [39,40].

For the determination of immunomodulation parameters, our strategy was to vaccinate mice with two doses of the vaccine by the intramuscular route, with an interval of 7 days between them. Seven days after the last dose, the animals were challenged with *L.amazonensis* promastigotes. The animals were euthanized 18 h after infection, which corresponded to the peak of DTH. At the site of infection, mice vaccinated with LaAg + SAP showed the following: increased cell permeability; the accumulation of neutrophils, eosinophils, macrophages and lymphocytes and increased mRNA transcripts of iNOS and decreased mRNA transcripts of TGF-β (Appendix A).

It has already been described that besides DTH being correlated with protection and the increase of Th1 cytokines, it also enhances NO and reduces visceralization, indicating enduring protection against *L.infantum* infection [31]. Studies that assessed nitrite levels as a reflection of NO in the serum of vaccinated animals (LbAg + SAP or LbAg + SAP + SGE) demonstrated an increase compared to non-vaccinated groups. This finding corroborates our RT-qPCR data, which also indicated higher levels of mRNA for iNOS in samples from the footpad of vaccinated animals (Appendix A). Collectively, these findings suggest that vaccinated and partially protected groups also exhibit elevated levels of iNOS or nitrite [41].

A previous study investigated the modulatory response associated with the susceptibility of BALB/c mice. Pre-vaccination with LaAg sensitized these animals for the specific production of IL-10 and TGF-β. These mice received 25 µg of LaAg or PBS (intramuscularly) at seven-day intervals between the two doses. This increase in TGF-β production was indicated from the supernatant of the draining lymph node cells. However, when anti-TGF-β antibody neutralization was promoted during the LaAg vaccination, there was an improvement in susceptibility, demonstrating the role of TGF-β as a critical factor in disease promotion [9]. According to our data, there was a decrease in TGF-β mRNA transcripts in the LaAg + SAP group, supporting the findings of the study that TGF-β participation is important in disease immunomodulation.

Another study evaluated the mRNA expression of different cytokines crucial for protection or susceptibility to *Leishmania* infection (IL-12, IFN-γ, TNF-α, IL-4, IL-13, TGF-β and IL-10) in the dermis of dogs at several time points (1, 12, 24 and 48 h) after inoculation with the LbSAP vaccine or its components. As a result of these experiments, at 12 h, IL-12 expression in the LbSap group was higher than in the SAP group, and IL-10 expression was also higher in the LbSAP group compared to the SAP group. Additionally, positive correlations were observed between the mRNA expression of type I cytokines (IL-12, IFN-γ TNF-α) and type II cytokines in groups inoculated with separate vaccine components and the LbSAP vaccine itself, indicating that cytokines at the infection site play a pivotal role in regulating the immune response concerning protection or susceptibility [42].

In our study, the presence of DTH in the early hours post-challenge provided evidence and confirmation that the cellular population in the microenvironment appears to indeed be associated with protection or susceptibility in the vaccine. In the popliteal lymph nodes, FACS labeling revealed that this increase in cell numbers was accompanied by increased numbers of both CD4^+^ and CD8^+^ T cells, corroborating the immunogenicity data. Still in the draining lymph nodes, it was observed that there was a significant increase in IFN-γ-producing CD4^+^ T cells, and a more significant increase in IFN-γ-producing CD8^+^ T cells, which also corroborates the immunogenicity data. In addition, CD8^+^ T cells also produced IL-10 and IL-4, revealing a mixed Th1 and Th2 profile of these cells. In contrast, CD4^+^ T cells did not produce IL-4, suggesting a profile more compromised with the Th1 response, which would favor the elimination of the parasite. In addition, it was observed that there was an increase in the number of CD4^+^ T cells producing IL-10, which is also positive, since a regulatory profile is also important during leishmaniasis, as an uncontrolled Th1 response is also part of the pathogenesis of this disease. The IL-10 production by CD8^+^ T cells also corroborates the immunogenicity data; however, in the immunogenicity experiments, there was a reduction in the frequency and number of IL-4-producing CD8^+^ T cells, while in the immunomodulation experiments, an increase in the number of IL-4-producing CD8^+^ T cells was observed. This fact shows that the infection may be modulating the production of IL-4 by these cells.

A study that evaluated the expression of cytokines and chemokines in the dermis of dogs vaccinated with LbSAP (Lb, SAP or both together) demonstrated an increase in the mRNA levels of cytokines, specifically IL-10, at 12 and 24 h. Additionally, positive correlations were observed between various cytokine expressions, including IFN-γ and TGF-β [42], suggesting that a diverse cytokine microenvironment developed following immunization with the vaccine. Post-vaccination, all dogs were challenged with *L.infantum* promastigotes and monitored bi-monthly through bone marrow aspirates and blood tests for immune response evaluation. Vaccinated dogs exhibited an enhanced immune response, indicating the potential efficacy of the LbAg vaccine [43], similar to the increased cytokine production observed with the LaAg + SAP vaccine.

Previously, our group tested MPLA/AddaVax™ plus LaAg intramuscularly with challenge with *L.amazonensis* (2 × 10^5^ or 2 × 10^6^) reduced lesion growth only at the peak and only with the lower parasite dose, with no reduction in parasite load. Mice challenged with 2×10^6^ were not protected by LaAg. Regardless of the parasite challenge, the combination of LaAg with MPLA/AddaVax™ did not significantly increase protection compared to LaAg alone. MPLA/AddaVax™ was not effective in enhancing the efficacy of the LaAg vaccine against cutaneous leishmaniasis [12].

The association of LaAg + SAP, by intramuscular route, reversed the non-protective effect of LaAg alone, reducing the lesion size and parasite burden of previously vaccinated BALB/c mice when compared to the PBS, LaAg and SAP groups, which shows a beneficial effect of using SAP as an adjuvant in the vaccine formulation. The protection mechanism is mainly based on the response of IFN-γ -producing CD4^+^ and CD8^+^ T cells in the popliteal lymph nodes. At the site of infection, it was observed that there was increased cell permeability, the accumulation of neutrophils, eosinophils, macrophages and lymphocytes and increased mRNA transcripts of iNOS and decreased mRNA transcripts of TGF-β. Altogether, for a deleterious antigen as LaAg, saponin works as a good adjuvant converting a non-protective vaccine into a protective vaccine.

## 5. Conclusions

The results indicate that the combination of LaAg with the SAP adjuvant provides significant protection against *Leishmania* infections, reducing lesions and parasite load in BALB/c mice. The LaAg + SAP vaccine induced a robust and protective immune response, with increased IFN-γ-producing T cells and reduced TGF-β, demonstrating its promising potential for controlling leishmaniasis.

## Figures and Tables

**Figure 1 vaccines-13-00129-f001:**
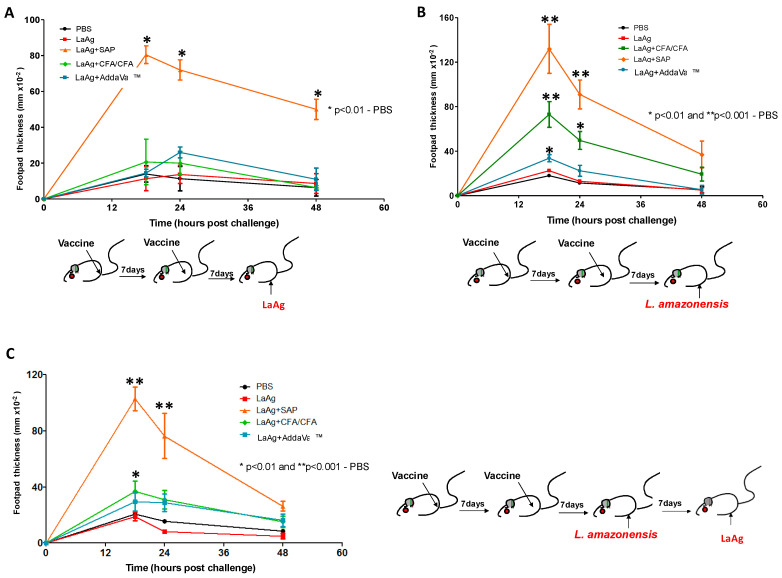
DTH response in BALB/c immunized with LaAg and different adjuvants. BALB/c mice were vaccinated twice, with an interval of one week between doses, intramuscularly with 100 μg of LaAg alone or associated with different adjuvants, namely CFA (80 µL/dose), SAP (100 μg/dose) or AddaVax^TM^ (50 µL/dose); in the control group, 100 μL of PBS was used. One week after the last vaccine dose, they were challenged in the footpad with LaAg, *L.amazonensis* or *L.amazonensis* and LaAg. Swelling was measured at times of 0, 18, 24 and 48 h after challenge. (**A**) Challenge with 100 μg LaAg. (**B**) Challenged with live *L.amazonensis* in stationary phase (2 × 10^6^ promastigotes). (**C**) Challenged with live *L.amazonensis* in stationary phase (2 × 10^6^ promastigotes) and 1 week after new challenge with 10 μg of LaAg. These results refer to one independent experiment and each group consisted of five animals. Results were expressed as mean ± SD (*n* = 5). * *p* < 0.05 and ** *p* < 0.01 in relation to the PBS group.

**Figure 2 vaccines-13-00129-f002:**
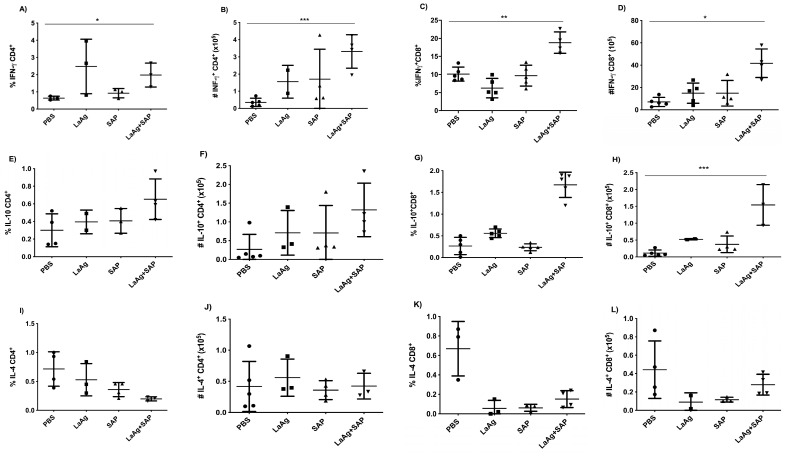
Phenotypic analysis of IFN-γ-, IL-10- and IL-4-producing CD4^+^ and CD8^+^ T cells in BALB/c mice vaccinated with LaAg + SAP. BALB/c mice were vaccinated twice intramuscularly, with an interval of 1 week between doses, and euthanized 3 days after the second vaccine dose. Phenotypic analysis of popliteal lymph nodes was carried out. Frequency (**A**) and absolute number (**B**) of IFN-γ-producing CD4^+^ cells. Frequency (**C**) and absolute number (**D**) of IFN-γ-producing CD8^+^ cells. Frequency (**E**) and absolute number (**F**) of IL-10-producing CD4^+^ cells. Frequency (**G**) and absolute number (**H**) of IL-10-producing CD8^+^ cells. Frequency (**I**) and absolute number (**J**) of IL-4-producing CD4^+^ cells. Frequency (**K**) and absolute number (**L**) of IL-4-producing CD8^+^ cells. These results refer to one independent experiment, with five animals per group. Results were obtained using FlowJo software and expressed as mean ± SD (*n* = 5). * *p* < 0.05, ** *p* < 0.01 and *** *p* < 0.001 in relation to the PBS group.

**Figure 3 vaccines-13-00129-f003:**
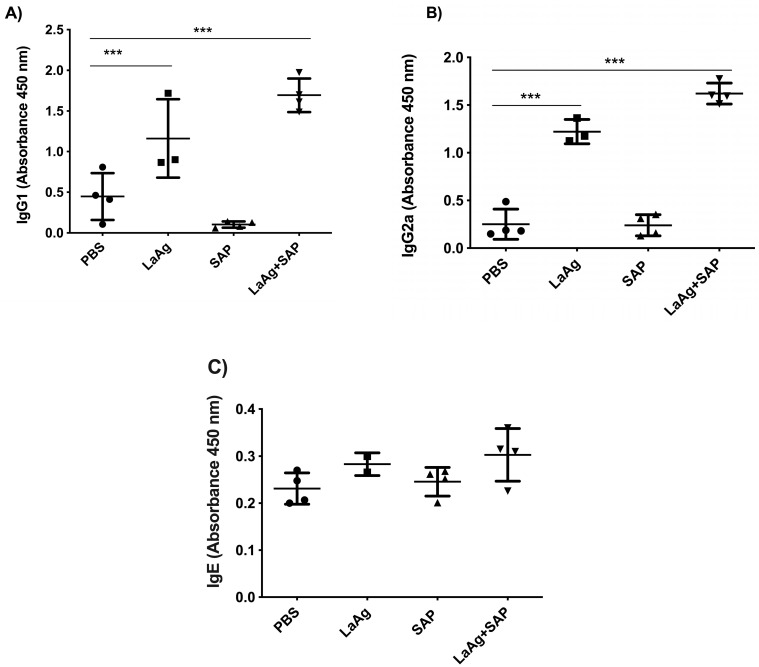
Antibody levels increased in mice vaccinated with LaAg + SAP. BALB/c mice were vaccinated twice intramuscularly, with an interval of 1 week between doses and euthanization 3 days after the second vaccine dose. The levels of IgG1, IgG2a and IgE antibodies in the plasma of the animals were determined. (**A**) IgG1 antibody level. (**B**) IgG2a antibody level. (**C**) IgE antibody level. This result refers to one independent experiment, with five animals per group. Results were expressed as mean ± SD. *** *p* < 0.001 in relation to the PBS group.

**Figure 4 vaccines-13-00129-f004:**
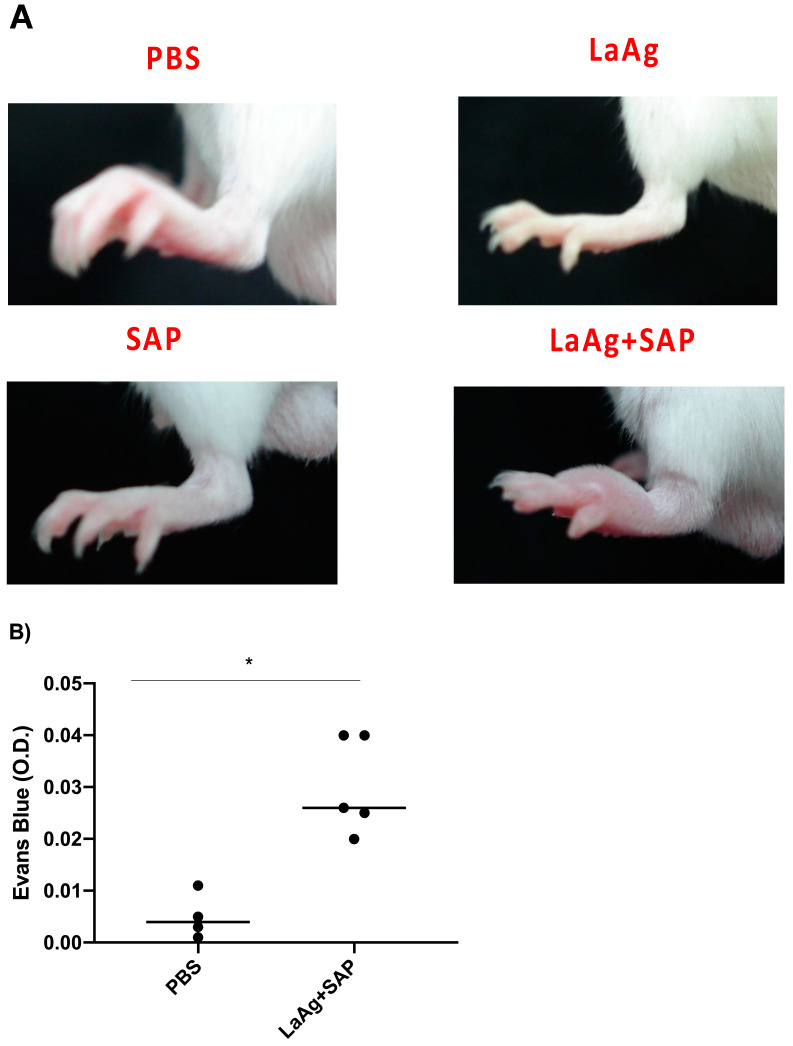
Cutaneous DTH and vascular permeability in BALB/c mice previously immunized with LaAg + SAP and their immunogenic controls, and after being challenged with *L.amazonensis*. (**A**) Illustrative photographs of the representative footpads of each group (there was no distance parameter control for photography) 18 h after challenge. These results refer to three independent experiments, with five animals per group. (**B**) Optical density (O.D.) of Evans blue dye. The data represent the subtraction of the O.D. of infected footpads to uninfected footpads. These data refer to one independent experiment, with three animals per group. All these results were expressed as mean ± SD. * *p* < 0.05 in relation to the PBS group.

**Figure 5 vaccines-13-00129-f005:**
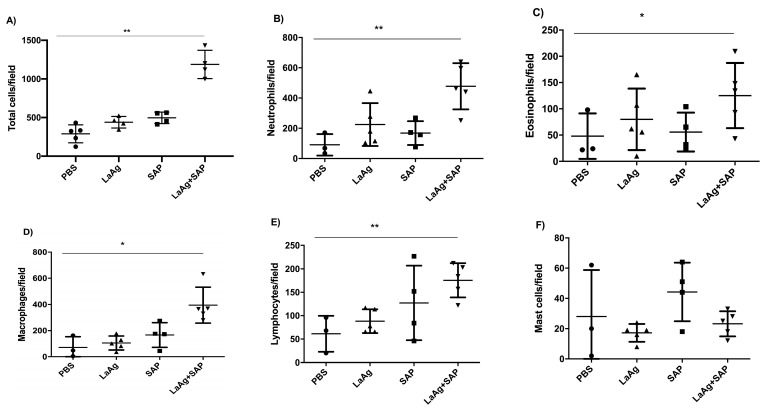
Histology of the footpads of BALB/c mice vaccinated with LaAg + SAP and challenged with *L.amazonensis*. BALB/c mice were immunized, infected with *L.amazonensis* and 18 h after challenge, the animals were euthanized. The infected footpads were fixed in paraformaldehyde and decalcified with EDTA. Sections of 4 μm were made followed by hematoxylin and eosin staining for counting. (**A**) Total cell count. (**B**) Neutrophils/field. (**C**) Eosinophils/field. (**D**) Macrophages/field. (**E**) Lymphocytes/field. (**F**) Mast cells/field. These results refer to one independent experiment, with five animals per group. The results were expressed as mean ± SD (*n* = 5). * *p* < 0.05 and ** *p* < 0.01 in relation to the PBS group.

**Figure 6 vaccines-13-00129-f006:**
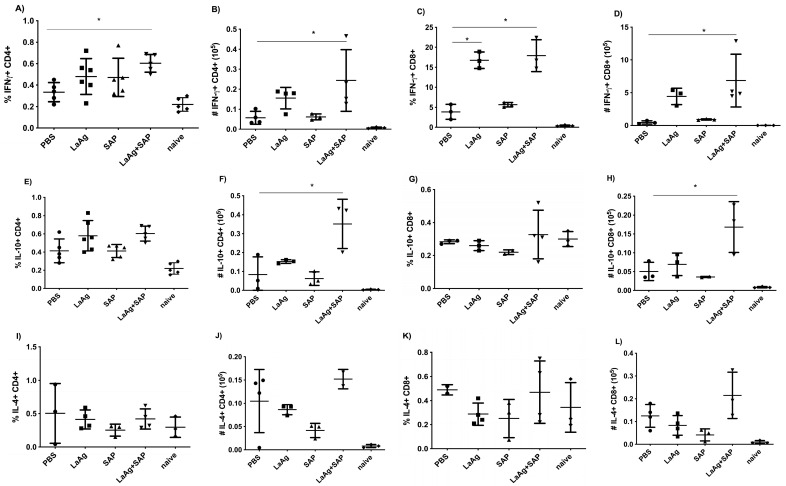
Phenotypic analysis of IFN-γ-, IL-10- and IL-4-producing CD4^+^ and CD8^+^ T cells in BALB/c mice vaccinated with LaAg + SAP and challenged with *L.amazonensis*. BALB/c mice were vaccinated twice intramuscularly, with an interval of one week between doses, challenged with *L.amazonensis* seven days after the last dose and euthanized 18 h after challenge. Phenotypic analysis of the popliteal lymph nodes was carried out. Frequency (**A**) and absolute number (**B**) of IFN-γ-producing CD4^+^ cells. Frequency (**C**) and absolute number (**D**) of IFN-γ-producing CD8^+^ cells. Frequency (**E**) and absolute number (**F**) of IL-10-producing CD4^+^ cells. Frequency (**G**) and absolute number (**H**) of IL-10-producing CD8^+^ cells. Frequency (**I**) and absolute number (**J**) of IL-4-producing CD4^+^ cells. Frequency (**K**) and absolute number (**L**) of IL-4-producing CD8^+^ cells. These results refer to one independent experiment, with five animals per group. Results were obtained using the FlowJo software and expressed as mean ± SD (*n* = 5). * *p* < 0.05 in relation to the PBS group.

**Figure 7 vaccines-13-00129-f007:**
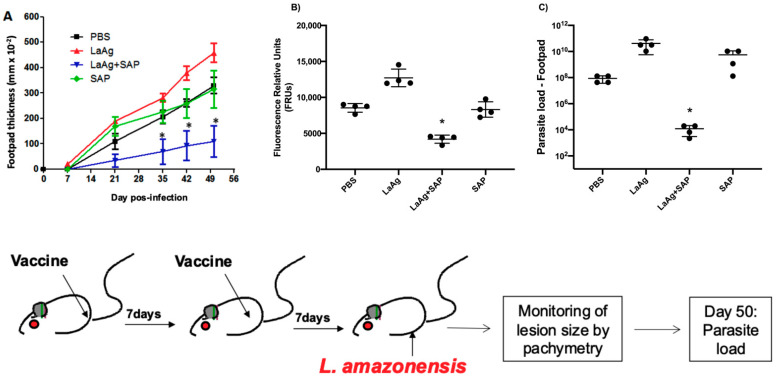
Efficacy of the LaAg + SAP vaccine in protecting against *L.amazonensis* infection in BALB/c mice. (**A**) Lesion size that was followed for 50 days after infection. Footpad thicknesses were monitored weekly by pachymetry. (**B**) Parasite load on day 50 after infection; the result is expressed as the relative fluorescence units from macerated footpads infected with *L.amazonensis* (GFP). (**C**) limited dilution assay analysis of parasite load. These results refer to one independent experiment, with five animals per group. Results were expressed as mean ± SD (*n* = 5). * *p* < 0.05 in relation to the PBS group.

## Data Availability

The original contributions presented in the study are included in the article. Further inquiries can be directed to the corresponding author.

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
