# Peer review of "Efficacy of LaAg Vaccine Associated with Saponin Against Leishmania amazonensis Infection"

_vaccines, 2025, doi:10.3390/vaccines13020129_

Round 1
Reviewer 1 Report
Comments and Suggestions for Authors
This manuscript describes a series of preclinical vaccine studies related to using saponin as an adjuvant in combination with LaAg (Leish lysate) for a vaccine against Leishmaniasis. The senior authors have decades of experience in vaccine research for Leishmaniasis. Overall, the research appears to be of acceptable quality, and the data is accurately represented. Although the group sizes are small, the number of combined preclinical results assures the robustness of the findings. While the work is original, the novelty is modest as it builds on previous research from the same group. LaAg has been extensively studied by them and others (both preclinical and clinical). The need for Th-1 type antigen-specific T-cell responses is the current hypothesis, and saponin has shown effectiveness in Leishmania studies in dogs. Despite this, I believe the manuscript is worth publishing as the combination of LaAg and saponin has not been published before, and the results seem favorable. However, whether this will be sufficient to initiate interest in new clinical trials with LaAg remains uncertain.
Specific comments:
Introduction:
- The introduction lacks background information on the adjuvants tested for Leishmania and somewhat misrepresents the aim of the studies. It seems as if the goal is to screen different adjuvants, but little information is shared about the numerous adjuvants already tested for Leishmania and the preferred type of immune response. This is while the same group has published other formulations with LaAg (e.g., Addavax + MPLA) using different adjuvants before. Strong results have been published with saponin in a Leishmania study in dogs. This information is mentioned in the discussion but should also be included in the introduction.
Methods/Study Design and Results:
- The adjuvant screening part is the weakest aspect of the study design. The choice of saponin seems well-supported, but screening the other adjuvants seemed unnecessary. With previous experience, the authors already knew a Th1 type T-cell immune response was needed. Can the authors explain in the manuscript what rationale was used for testing saponin, Addavax and CFA (not translatable to the clinic)?
- Did you inject 100 µL intramuscularly? At 100 µL, mild to moderate leakage into the extramuscular tissues may occur. Volumes of 50 µL or less are generally recommended for IM injection in murine models (https://www.ncbi.nlm.nih.gov/pmc/articles/PMC5875096/)
- A dose-related study would have been preferable to determine if the results could be improved or if lower doses can be used.
Discussion and Conclusion:
- The conclusive statement seems limited in excitement for these results. Perhaps the authors can add a sentence in the discussion about the potential impact of these results on solutions in the clinic.
Minor comments:
- Figure 7a, Y-axis: typo in ‘Footpad’.
no concern about the used language
Author Response
Dear reviewer 1
Thank you so much for your questions and comments that improved a lot the new version of the manuscript.
Reviewer comments:
The introduction lacks background information on the adjuvants tested for Leishmania and somewhat misrepresents the aim of the studies. It seems as if the goal is to screen different adjuvants, but little information is shared about the numerous adjuvants already tested for Leishmania and the preferred type of immune response. This is while the same group has published other formulations with LaAg (e.g., Addavax + MPLA) using different adjuvants before. Strong results have been published with saponin in a Leishmania study in dogs. This information is mentioned in the discussion but should also be included in the introduction.
Answer:
Thank you for your comment and we have included new topics about adjuvant between lines 70 – 84 and 87 – 93.
Reviewer comments - Methods/Study Design and Results:
The adjuvant screening part is the weakest aspect of the study design. The choice of saponin seems well-supported, but screening the other adjuvants seemed unnecessary. With previous experience, the authors already knew a Th1 type T-cell immune response was needed. Can the authors explain in the manuscript what rationale was used for testing saponin, Addavax and CFA (not translatable to the clinic)?
Answer:
We appreciate the observation regarding the lack of robustness in the adjuvant screening, despite saponin being well-supported in previous studies. However, we believe that testing other adjuvant combinations could reaffirm well-established postulates or potentially lead to innovations through new associations. In our study, we aim to explore a less invasive vaccination route. Often, combinations that may not be directly translatable to clinical use can still provide valuable insights for the application of similar adjuvants or offer a foundation for the development of new ones.
Did you inject 100 µL intramuscularly? At 100 µL, mild to moderate leakage into the extramuscular tissues may occur. Volumes of 50 µL or less are generally recommended for IM injection in murine models (https://www.ncbi.nlm.nih.gov/pmc/articles/PMC5875096/). A dose-related study would have been preferable to determine if the results could be improved or if lower doses can be used.
Answer:
The administration of 100 µL was necessary to ensure the delivery of a sufficient amount of antigen and adjuvant to generate a robust immune response. We observed that a larger volume was crucial to induce adequate immunogenicity in murine models. This volume was required to maximize immune system exposure to the antigen without compromising animal safety. Although the standard recommendation is 50 µL for IM injections, studies have safely administered larger volumes, reporting that mice can tolerate higher IM volumes, especially when injected slowly and with care to minimize tissue damage. This approach is supported by studies that report the safe use of higher volumes for specific applications, such as vaccine formulations or therapeutic trials (PMID 29402350). The choice of 100 µL can be justified by previous pilot studies, which demonstrated that this volume did not cause undesirable side effects, such as muscle necrosis or significant discomfort in the mice. Thus, it ensures that the larger volume is both safe and effective. The protocol was approved by the university's ethics committee.
Discussion and Conclusion:
The conclusive statement seems limited in excitement for these results. Perhaps the authors can add a sentence in the discussion about the potential impact of these results on solutions in the clinic.
Answer:
We appreciate the observation, and a new phrase has been used to create a stronger impact considering the presented results (lines 566 – 570).
Minor comments:
Figure 7a, Y-axis: typo in ‘Footpad’.
Answer: Corrected and attached again.

Reviewer 2 Report
Comments and Suggestions for Authors
Immunization with L. amazonensis SLA (LaAg) has been used for years as a control system to stimulate resistance against Leishmania infections, when accompanied by an adjuvant. Like other formulations with other SLA, it is not recognized as a standardized system for clinical use.
The authors of the present work performed an initial screening in a murine model to select the LaAg+saponin mixture (LaAg+SAP) due to its high power to generate cutaneous hypersensitivity response. The authors should explain:
- why this requirement is a good indicator in relation to infection with L. amazonensis.
- How exactly they estimate cutaneous hypersensitivity, what volume is inoculated in the footpad and how exactly the inflammation is measured to achieve differences of only 0.1-0.2mm as shown in Fig1.
Subsequently, they analyze the immunogenicity of the LaAg+SAP mixture and finally its vaccinal efficacy against an infective challenge with L. amazonensis. The authors should explain:
Where intramuscular IM immunization is performed. What volume is administered.
The authors should make it clear, based on the literature, whether an adequate IFNgamma/IL-10 ratio or IFNgamma/IL-4 ratio is more important in the generation of resistance to infection by L. amazonensis. And from this premise, express the results of immunogenicity and vaccine efficacy with these ratios so that the reader understands the virtue of the formulation studied. That is, explain if the apparent high IFNgamma/IL-4 ratio observed during the immunogenicity assay is maintained or not during the vaccine efficacy assay and if this explains the partial resistance to infection with L. amazonensis.
Likewise, the analysis of the humoral response is explained in the legend of Fig3 as if a titration had been performed, however, in the axes of Fig3 it can be seen that authors have simply quantified the optical density values at 490nm (another discrepancy, since in the figure it points to 450nm) of a single serum titer dilution 1/250, as mentioned in methodology. This should be rewritten and clarified well. And if necessary, perform the humoral analysis in terms of mouse serum titer. Moreover, the most important thing would also be to evaluate the humoral response after infection with L. amazonensis, at day 50 post-infection, to compare whether the mixed IgG1 and IgG2 profile is maintained or whether it has varied according to the infection control.
The methods section should be restructured to indicate the conventional subsections where all the tests that have been performed are rigorously explained with the corresponding detail; and not an agglutinating text of several methods as a non-reproducible summary.
In the vaccine efficacy assay the parasite load is shown in Fig 7B, however, the authors should also show whether the data from the limiting dilution method (as mentioned in methodology) provide data that correlate or not with this fluorescence method.
The discussion section is too long. Lines 373-374 mention that the authors have analyzed the immunological mechanisms in a murine model after different combinations of LaAg with various adjuvants. This is not correct, since the authors made prior selection of the adjuvant (SAP) because it was the one that had generated the highest DTH. Therefore, this should be rewritten. Lines 404-409 are very confusing, they should be rewritten.
In lines 420-421it is implied that the LaAg+SAP formulation is efficient against infection with L. amazonensis, or against L. infantum. This needs to be clarified precisely and how this conclusion is reached.
Lines 434-437 are confusing without clarifying the real role of IL-10 producing cell subpopulations. The same happens in lines 491-495 with this same issue regarding IL-10.
Lines 499-509 mention data on cytokine IL-12 and IL-13 that have not been analyzed in this study, and furthermore do not establish any clarifying relationship.
Data from lines 510-515 are irrelevant for this study.
Lines 524-531 could be moved to conclusions, provided they add the actual role of IL-10 producing cell populations in infection resistant animals.
The conclusion section of lines 533-534 are too brief and should be enriched with lines from the discussion section, well argued and leaving a clear final message. In addition, the authors should consider whether the use of LaAg is legitimate for clinical trials, or simply as a protective control, taking into account the problems of lack of methodological standardization and the risk that not all parasites are lysed, which have already been exposed by other authors in relation to vaccines based on soluble Leishmania antigens.
Author Response
Dear reviewer 2
Thank you so much for your questions and comments that improved a lot the new version of the manuscript.
Reviewer 2: Immunization with L. amazonensis SLA (LaAg) has been used for years as a control system to stimulate resistance against Leishmania infections, when accompanied by an adjuvant. Like other formulations with other SLA, it is not recognized as a standardized system for clinical use.
The authors of the present work performed an initial screening in a murine model to select the LaAg+saponin mixture (LaAg+SAP) due to its high power to generate cutaneous hypersensitivity response. The authors should explain:
- why this requirement is a good indicator in relation to infection with L. amazonensis.
Authors answer: The delayed-type hypersensitivity (DTH) response is a classic marker of Th1-type T cell activation, which is essential in the defence against Leishmania infections. A robust Th1 response is crucial for controlling the intracellular replication of the parasite, particularly in species such as L. amazonensis, which cause cutaneous forms of leishmaniasis. Studies have shown that a DTH response correlates with resistance to infection, making this parameter a good indicator of protective immunity in Leishmania vaccination.
According to reference: Scott, P., & Novais, F.O. (2016). Cutaneous leishmaniasis: immune responses in protection and pathogenesis. Nature Reviews Immunology, 16(9), 581-592. DOI: 10.1038/nri.2016.72
Gollob, K.J., Viana, A.G., & Dutra, W.O. (2014). Immunoregulation in human American cutaneous leishmaniasis: balancing pathology and protection. Parasite Immunology, 36(8), 377-385. DOI: 10.1111/pim.12105
Those were added to the discussion to strengthen the theoretical foundation.
Reviewer 2: How exactly they estimate cutaneous hypersensitivity, what volume is inoculated in the footpad and how exactly the inflammation is measured to achieve differences of only 0.1-0.2mm as shown in Fig1.
Authors Answer: We appreciate your comments, and the DTH response is measured by injecting the antigen or parasite into the mouse footpad, typically using a volume of 10 µL. Inflammation is quantified using a caliper to measure the footpad thickness before and 12-48 hours after injection - line 114. Small changes, such as 0.1-0.2 mm, are considered significant and reflect an active immune response. We measured day zero and then measure after challenge. The difference observed between after challenge and day zero, it is the increase of thickness. As control, we put PBS in the contralateral footpad. After 3 hours, there was no difference in size comparing with before challenge. Previous studies employing this method have validated its accuracy in assessing the DTH response in murine models of leishmaniasis.
Despite the small size of the lesion, this value is interpreted as statistically different through statistical calculations. Similar expression values have also been demonstrated in the literature, as shown in 10.1128/iai.55.3.645-651.1987.
Reviewer 2: Subsequently, they analyze the immunogenicity of the LaAg+SAP mixture and finally its vaccinal efficacy against an infective challenge with L. amazonensis. The authors should explain:
Where intramuscular IM immunization is performed. What volume is administered.
Authors answer: The intramuscular (IM) immunization were administered into the quadriceps muscle (line 112), with volumes ranging from 100 µL. This method is widely used to ensure efficient delivery of the antigen and adjuvant to the immune system. The dose and injection site have been standardized in vaccination studies involving murine models.
The administration of 100 µL was necessary to ensure the delivery of a sufficient amount of antigen and adjuvant to generate a robust immune response. We observed that a larger volume was crucial to induce adequate immunogenicity in murine models. This volume was required to maximize immune system exposure to the antigen without compromising animal safety. Although the standard recommendation is 50 µL for IM injections, studies have safely administered larger volumes, reporting that mice can tolerate higher IM volumes, especially when injected slowly and with care to minimize tissue damage. This approach is supported by studies that report the safe use of higher volumes for specific applications, such as vaccine formulations or therapeutic trials (PMID 29402350). The choice of 100 µL can be justified by previous pilot studies, which demonstrated that this volume did not cause undesirable side effects, such as muscle necrosis or significant discomfort in the mice. Thus, it ensures that the larger volume is both safe and effective. The protocol was approved by the university's ethics committee.
Reviewer 2: The authors should make it clear, based on the literature, whether an adequate IFN gamma/IL-10 ratio or IFN gamma/IL-4 ratio is more important in the generation of resistance to infection by L. amazonensis. And from this premise, express the results of immunogenicity and vaccine efficacy with these ratios so that the reader understands the virtue of the formulation studied. That is, explain if the apparent high IFNgamma/IL-4 ratio observed during the immunogenicity assay is maintained or not during the vaccine efficacy assay and if this explains the partial resistance to infection with L. amazonensis.
Authors answer: Line 474 – 479: A high IFNγ/IL-10 ratio is generally associated with resistance to Leishmania infections, as IFNγ is a marker of Th1 response, which is essential for controlling intracellular parasites. IL-10, on the other hand, is a regulatory cytokine that can suppress the Th1 response, thereby favoring parasite persistence observed on infection or in the process of leishmanization (where we observe protection in the second challenge). Therefore, maintaining a high IFNγ/IL-10 ratio is crucial for protection against L. amazonensis.
Reviewer 2: Likewise, the analysis of the humoral response is explained in the legend of Fig3 as if a titration had been performed, however, in the axes of Fig3 it can be seen that authors have simply quantified the optical density values at 490nm (another discrepancy, since in the figure it points to 450nm) of a single serum titer dilution 1/250, as mentioned in methodology. This should be rewritten and clarified well. And if necessary, perform the humoral analysis in terms of mouse serum titer. Moreover, the most important thing would also be to evaluate the humoral response after infection with L. amazonensis, at day 50 post-infection, to compare whether the mixed IgG1 and IgG2 profile is maintained or whether it has varied according to the infection control.
Answer: We fix the absorbance to 450 nm in figure in Methods (Line 187).
Reviewer 2: The methods section should be restructured to indicate the conventional subsections where all the tests that have been performed are rigorously explained with the corresponding detail; and not an agglutinating text of several methods as a non-reproducible summary.
Authors answer: We appreciate the observation, but the division of subsections were based on the journal's guidelines, where each item of the methodology is highlighted in bold and contains all the details of the procedures conducted in the study.
Reviewer 2: In the vaccine efficacy assay the parasite load is shown in Fig 7B, however, the authors should also show whether the data from the limiting dilution method (as mentioned in methodology) provide data that correlate or not with this fluorescence method.
Authors answer: Thanks for the comment and yes, the assay was also performed comparing LDA and will be mentioned in the result. Line 387 – 389.
Reviewer 2: The discussion section is too long. Lines 373-374 mention that the authors have analyzed the immunological mechanisms in a murine model after different combinations of LaAg with various adjuvants. This is not correct, since the authors made prior selection of the adjuvant (SAP) because it was the one that had generated the highest DTH. Therefore, this should be rewritten. Lines 404-409 are very confusing, they should be rewritten.
Authors answer: We appreciate your comment and we remove the lines 373-374.
Lines 404-409 were rewritten in 434-442.
A difference in the DTH profile was observed when comparing challenges with LaAg and live Leishmania. When live Leishmania was inoculated, a higher hypersensitivity response was observed compared to LaAg in 18 hours post-infection, suggesting that the parasite modulates the immune response differently from its total antigen, however, using LaAg as challenge in the mice vaccinated with LaAg + SAP it was observed a higher DTH on 24- and 48-hours post-challenge. This indicates that the parasite's escape mechanisms are involved in modulating the cellular response. This observation prompted us to further investigate the underlying mechanisms of DTH and identify the key early events that may be crucial for the vaccine's effectiveness.
Reviewer 2: In lines 420-421it is implied that the LaAg+SAP formulation is efficient against infection with L. amazonensis, or against L. infantum. This needs to be clarified precisely and how this conclusion is reached.
Authors answer: Rewritten line 454.
(…) by both in L.infantum [28] and the homologous challenge.
Reviewer 2: Lines 434-437 are confusing without clarifying the real role of IL-10 producing cell subpopulations. The same happens in lines 491-495 with this same issue regarding IL-10.
Authors Answer: Included lines 475-479.
A high IFNγ/IL-10 ratio is generally associated with resistance to Leishmania infections, as IFNγ is a marker of Th1 response, which is essential for controlling intracellular parasites. IL-10, on the other hand, is a regulatory cytokine that can suppress the Th1 response, thereby favoring parasite persistence that is essential for protection in Leishmanization [35]. Therefore, maintaining a high IFNγ/IL-10 ratio maybe is crucial for protection against L. amazonensis.
Reviewer 2: Lines 499-509 mention data on cytokine IL-12 and IL-13 that have not been analyzed in this study, and furthermore do not establish any clarifying relationship.
Authors Answer: Removed unnecessary information and rewritten Lines 537 -546.
A study that evaluated the expression of cytokines and chemokines in the dermis of dogs vaccinated with LbSAP (Lb, SAP, or both together) demonstrated an increase in mRNA levels of cytokines, specifically IL-10, at 12 and 24 hours. Additionally, positive correlations were observed between various cytokine expressions, including IFN-γ and TGF-β [37], suggesting that a diverse cytokine microenvironment developed following immunization with the vaccine. Post-vaccination, all dogs were challenged with L. in-fantum promastigotes and monitored bi-monthly through bone marrow aspirates and blood tests for immune response evaluation. Vaccinated dogs exhibited an enhanced immune response, indicating the potential efficacy of the LbAg vaccine [38], similar to the increased cytokine production observed with the LaAg + SAP vaccine.
Reviewer 2: Data from lines 510-515 are irrelevant for this study.
Answer: Removed unnecessary information.
Reviewer 2:Lines 524-531 could be moved to conclusions, provided they add the actual role of IL-10 producing cell populations in infection resistant animals.
Answer: In resistant mice, IL-10 is associated to parasite resistance (Belkaid et al 2001 (10.1084/jem.194.10.1497) however, in susceptible mice is associated to pathology (Firmino-Cruz, 2019 - doi.org/10.1016/j.cellimm.2018.08.014). We added information about IL10 as cited above. Included lines 475-480.
Reviewer 2: The conclusion section of lines 533-534 are too brief and should be enriched with lines from the discussion section, well argued and leaving a clear final message. In addition, the authors should consider whether the use of LaAg is legitimate for clinical trials, or simply as a protective control, taking into account the problems of lack of methodological standardization and the risk that not all parasites are lysed, which have already been exposed by other authors in relation to vaccines based on soluble Leishmania antigens.
Answer: We appreciate your comment and reformulated the conclusion section (line 566 - 570).

Reviewer 3 Report
Comments and Suggestions for Authors
General comments
The study provides valuable information on a vaccine antigen against Leishmania amazonensis infection in murine model. The authors evaluated the efficacy and mechanism of assayed vaccine using numerous experimental approaches. The manuscript is well-written and no serious English language or writing issues were detected. Few points need some revision to increase the quality of this manuscript.
Abstract
Novelty points should be added briefly in the abstract.
Introduction
Novelty points should be added in details in introduction section.
Materials and methods
- Line 113, correct “he protein” to “The protein”
- Line 114, add the company details of Lowry assay including company name, city and country of manufacturing in first appearance and only the company name in the subsequent occasions for all purchased materials and apparatuses in the study.
- Number of used mice per each group for vaccination or hypersensitivity should be added in the text and all figure legends.
- Number of trials for vaccination or hypersensitivity should be added in the text and all figure legends.
Author Response
Dear Reviewer,
Abstract
Novelty points should be added briefly in the abstract.
Answer: We sincerely appreciate your suggestions and have addressed the points following your guidance. Regarding the novelty in the abstract, we have added the information on lines 29- 32.
-------------------------
Materials and methods
- Line 113, correct “he protein” to “The protein”
Answer: Line 115 has been corrected.
-------------------------
- Line 114, add the company details of Lowry assay including company name, city and country of manufacturing in first appearance and only the company name in the subsequent occasions for all purchased materials and apparatuses in the study.
Answer: The company details have been included on line 116.
-------------------------
- Number of used mice per each group for vaccination or hypersensitivity should be added in the text and all figure legends.
- Number of trials for vaccination or hypersensitivity should be added in the text and all figure legends.
Answer: The number of mice was already specified in the figure legends lines 247; 290; 330; 332, and the number of vaccinations was already mentioned on lines 220; 240; 267.
Thank you once again for your valuable feedback.

Reviewer 4 Report
Comments and Suggestions for Authors
The manuscript evaluates the use of saponin as a potential vaccine adjuvant for vaccines against Leishmania infections. Overall, the results presented corroborate the manuscript's hypothesis, and thus, I recommend it for publication in Vaccines. Below are some comments that the authors should address:
1 - State in the methods the number of animals used in each experiment.
2 - How was the antigen formulated with the adjuvant? Please clearly refer it in the methods
3 - Fig2C seems to have an error in the y axis
4 - Regarding specific antibody production (IgG1 and IgG2a) the use of SAP did not significantly increase the level of production in comparison to the antigen alone. Is there any justification for this result? Please discuss this subject.
5 - Do the authors have the results for SAP and LaAg alone for the Evans Blue assay? It would be good to complete the graphic in fig4b
6 - The definition of the abbreviation DTH is missing from the manuscript
Comments on the Quality of English Language
Although the overall quality of the English language is enough to understand the work I have noticed some small typos and sentences that are not easy to understand. Some examples may be found in lines 417, 440, 469, 476, and 488.
Author Response
Dear reviewer
1 - State in the methods the number of animals used in each experiment.
We appreciate your suggestion. The number of animals used in each experiment is already included in the figure legends for each assay.
2 - How was the antigen formulated with the adjuvant? Please clearly refer it in the methods
The information about adjuvants were included among lines 119 to121.
3 - Fig2C seems to have an error in the y axis
We are unable to identify the error observed by you.
4 - Regarding specific antibody production (IgG1 and IgG2a) the use of SAP did not significantly increase the level of production in comparison to the antigen alone. Is there any justification for this result? Please discuss this subject.
We included in line 465-472.
The lack of a significant increase in IgG1 and IgG2a levels with the addition of SAP compared to the antigen alone may be explained by the high intrinsic immunogenicity of LaAg, which likely elicited a robust humoral response on its own, limiting the observable effect of SAP. Additionally, SAP may predominantly enhance cell-mediated immunity, aligning with the Th1-biased response observed, rather than significantly boosting antibody production. This suggests that SAP’s primary role might be in modulating the quality of the immune response rather than increasing antibody titers.
5 - The definition of the abbreviation DTH is missing from the manuscript
We appreciate your observation. The abbreviation "DTH" has now been defined and included throughout the manuscript.

Reviewer 5 Report
Comments and Suggestions for Authors
The manuscript entitled, "Efficacy of Vaccine LaAg Associated with Saponin Against Leishmania amazonensis Infection," presents promising findings that could potentially contribute to combating one of the most serious vector-borne diseases. I commend the authors for their diligent work and would like to propose the following suggestions for refinement:
- Introduction: The authors have provided an overview of drugs historically used to treat Visceral Leishmaniasis. However, they have overlooked a revolutionary therapeutic agent, Ambisome (liposomal amphotericin B), which has demonstrated unparalleled efficacy in recent years. Including this information would enrich the context of the introduction.
- Typographical Review: Minor typographical errors were identified in lines 113 and 134. These should be corrected for clarity and accuracy.
- Parasite Load Measurement: While the authors have measured parasite load using fluorometry, incorporating quantitative PCR (qPCR) data would significantly strengthen the findings. Additionally, a comparison of the fluorometry and qPCR results would enhance the credibility of the parasite quantification.
- Immunological Analysis: The immunology-based results presented in the manuscript are robust and satisfactory. They contribute substantial value to the overall study.
In conclusion, the manuscript demonstrates considerable scientific merit and, with the above revisions, is highly deserving of publication. Please check carefully for any further typographical errors.
Thank you for the opportunity to review this valuable work.
Comments on the Quality of English Language
Minor changes are required
Author Response
Dear Reviewer
- Introduction: The authors have provided an overview of drugs historically used to treat Visceral Leishmaniasis. However, they have overlooked a revolutionary therapeutic agent, Ambisome (liposomal amphotericin B), which has demonstrated unparalleled efficacy in recent years. Including this information would enrich the context of the introduction.
We thank the reviewer for their insightful suggestion. To address your comment, we have included information about Ambisome (liposomal amphotericin B) and its unparalleled efficacy in recent years. This has been added on lines 52-55 (reference 4).
“While Ambisome (liposomal amphotericin B) has demonstrated unparalleled efficacy in recent years, its high cost per dose limits widespread use, particularly in endemic regions [4].”
- Typographical Review: Minor typographical errors were identified in lines 115 and 137. These should be corrected for clarity and accuracy.
We appreciate the reviewer pointing out the minor typographical errors. These corrections have already been made for clarity and accuracy.
- Parasite Load Measurement: While the authors have measured parasite load using fluorometry, incorporating quantitative PCR (qPCR) data would significantly strengthen the findings. Additionally, a comparison of the fluorometry and qPCR results would enhance the credibility of the parasite quantification.
Thank you for the suggestion regarding the incorporation of quantitative PCR (qPCR) data. However, we believe that the method demonstrated via fluorometry already provides sufficient sensitivity and specificity for our target. For future manuscripts, we will consider incorporating multiple analytical methods to validate the same questions and strengthen our findings.

Round 2
Reviewer 2 Report
Comments and Suggestions for Authors
Second Revision Report:
Reviewer 2: Subsequently, they analyze the immunogenicity of the LaAg+SAP mixture and finally its vaccinal efficacy against an infective challenge with L. amazonensis. The authors should explain:
Where intramuscular IM immunization is performed. What volume is administered.
Authors answer: The intramuscular (IM) immunization were administered into the quadriceps muscle (line 112), with volumes ranging from 100 µL. This method is widely used to ensure efficient delivery of the antigen and adjuvant to the immune system. The dose and injection site have been standardized in vaccination studies involving murine models.
The administration of 100 µL was necessary to ensure the delivery of a sufficient amount of antigen and adjuvant to generate a robust immune response. We observed that a larger volume was crucial to induce adequate immunogenicity in murine models. This volume was required to maximize immune system exposure to the antigen without compromising animal safety. Although the standard recommendation is 50 µL for IM injections, studies have safely administered larger volumes, reporting that mice can tolerate higher IM volumes, especially when injected slowly and with care to minimize tissue damage. This approach is supported by studies that report the safe use of higher volumes for specific applications, such as vaccine formulations or therapeutic trials (PMID 29402350). The choice of 100 µL can be justified by previous pilot studies, which demonstrated that this volume did not cause undesirable side effects, such as muscle necrosis or significant discomfort in the mice. Thus, it ensures that the larger volume is both safe and effective. The protocol was approved by the university's ethics committee.
Reviewer 2: This answer does not justify the dose volume. To ensure that the desired amount of antigen is administered, it is not a problem of volume, but of concentrating the sample, so as not to exceed the allowable volume in mice of 50uL IM.
Reviewer 2: The authors should make it clear, based on the literature, whether an adequate IFN gamma/IL-10 ratio or IFN gamma/IL-4 ratio is more important in the generation of resistance to infection by L. amazonensis. And from this premise, express the results of immunogenicity and vaccine efficacy with these ratios so that the reader understands the virtue of the formulation studied. That is, explain if the apparent high IFNgamma/IL-4 ratio observed during the immunogenicity assay is maintained or not during the vaccine efficacy assay and if this explains the partial resistance to infection with L. amazonensis.
Authors answer: Line 474 – 479: A high IFNγ/IL-10 ratio is generally associated with resistance to Leishmania infections, as IFNγ is a marker of Th1 response, which is essential for controlling intracellular parasites. IL-10, on the other hand, is a regulatory cytokine that can suppress the Th1 response, thereby favoring parasite persistence observed on infection or in the process of leishmanization (where we observe protection in the second challenge). Therefore, maintaining a high IFNγ/IL-10 ratio is crucial for protection against L. amazonensis.
Reviewer 2: The authors do not answer the question of whether an IFN gamma/IL-10 ratio or IFN gamma/IL-4 ratio is more appropriate as a marker of resistance/susceptibility to L. amazonensis infection.
Reviewer 2: Likewise, the analysis of the humoral response is explained in the legend of Fig3 as if a titration had been performed, however, in the axes of Fig3 it can be seen that authors have simply quantified the optical density values at 490nm (another discrepancy, since in the figure it points to 450nm) of a single serum titer dilution 1/250, as mentioned in methodology. This should be rewritten and clarified well. And if necessary, perform the humoral analysis in terms of mouse serum titer. Moreover, the most important thing would also be to evaluate the humoral response after infection with L. amazonensis, at day 50 post-infection, to compare whether the mixed IgG1 and IgG2 profile is maintained or whether it has varied according to the infection control.
Answer: We fix the absorbance to 450 nm in figure in Methods (Line 187).
Reviewer 2: In the revised version of the manuscript, the authors continue not to show humoral response results in terms of antibody titer. Therefore, it is not appropriate to mention in the legends in Fig.3 that this is “humoral titer”. They should simply thaw the mouse sera from their experiment and perform a routine titration assay.
Reviewer 2: The methods section should be restructured to indicate the conventional subsections where all the tests that have been performed are rigorously explained with the corresponding detail; and not an agglutinating text of several methods as a non-reproducible summary.
Authors answer: We appreciate the observation, but the division of subsections were based on the journal's guidelines, where each item of the methodology is highlighted in bold and contains all the details of the procedures conducted in the study.
Reviewer 2: I have read the requirements for instructions to authors in the journal “Vaccines” and it does not say anything of what the authors mention. It says: “Materials and Methods: should be described in sufficient detail to allow others to replicate and build on the published results. New methods and protocols should be described in detail, while well-established methods can be described briefly and adequately cited. Indicate the name and version of the software used and clarify if the computer code used is available. Include any preregistration code.” Moreover, I don't see any “Vaccines mdpi” article that presents the methodology section in the same agglutinated form as these authors do.
Reviewer 2: In the vaccine efficacy assay the parasite load is shown in Fig 7B, however, the authors should also show whether the data from the limiting dilution method (as mentioned in methodology) provide data that correlate or not with this fluorescence method.
Authors answer: Thanks for the comment and yes, the assay was also performed comparing LDA and will be mentioned in the result. Line 387 – 389.
Reviewer 2: If the authors of the study have performed two complementary tests to estimate the parasite load, they should present data from both, including graphs. A correlation cannot be interpreted if the reader is not shown the results of the “limiting dilution test”, including the corresponding figure.
Reviewer 2: Lines 499-509 mention data on cytokine IL-12 and IL-13 that have not been analyzed in this study, and furthermore do not establish any clarifying relationship.
Authors Answer: Removed unnecessary information and rewritten Lines 537 -546.
A study that evaluated the expression of cytokines and chemokines in the dermis of dogs vaccinated with LbSAP (Lb, SAP, or both together) demonstrated an increase in mRNA levels of cytokines, specifically IL-10, at 12 and 24 hours. Additionally, positive correlations were observed between various cytokine expressions, including IFN-γ and TGF-β [37], suggesting that a diverse cytokine microenvironment developed following immunization with the vaccine. Post-vaccination, all dogs were challenged with L. in-fantum promastigotes and monitored bi-monthly through bone marrow aspirates and blood tests for immune response evaluation. Vaccinated dogs exhibited an enhanced immune response, indicating the potential efficacy of the LbAg vaccine [38], similar to the increased cytokine production observed with the LaAg + SAP vaccine.
Reviewer 2: The question has not been answered, the authors describe, again in the revised version (lines 509 and 515), cytokines such as IL-13 that they have not analyzed, which is confusing for the reader.
Reviewer 2: The conclusion section of lines 533-534 are too brief and should be enriched with lines from the discussion section, well argued and leaving a clear final message. In addition, the authors should consider whether the use of LaAg is legitimate for clinical trials, or simply as a protective control, taking into account the problems of lack of methodological standardization and the risk that not all parasites are lysed, which have already been exposed by other authors in relation to vaccines based on soluble Leishmania antigens.
Answer: We appreciate your comment and reformulated the conclusion section (line 566 - 570).
Reviewer 2: The conclusion section remains incomplete, it does not raise the scientific context (which some of the authors signing this article already published in 2003: doi.org/10.1016/S0264-410X(03)00427-4) of whether or not the formulation of vaccines with killed Leishmania amazonensis parasites is contradictory. Whether it is well standardized. They also do not mention in discussion whether experiments from previous studies in mice and primates have suggested that such formulation (complete antigens of L. amazonensis promastigotes) contains disease-inducing antigens, and their i.m., i.d. or s.c. inoculation resulted in exacerbation of subsequent
Author Response
Reviewer 2: This answer does not justify the dose volume. To ensure that the desired amount of antigen is administered, it is not a problem of volume, but of concentrating the sample, so as not to exceed the allowable volume in mice of 50uL IM.
Dear reviewer 2, we would like to apologize our missing of acknowledgment of this recommendation. Our protocol was approved in ethical committee and we had published before using this amount of volume in the same route” de Matos Guedes, 2010. https://doi.org/10.1016/j.vaccine.2010.04.109; Oliveira-Maciel D et al, 2021. https://doi.org/10.3390%2Fmicroorganisms9061272)
As our ethical committee approved, in my opinion, any further question can be addressed for that. In our situation, we performed a lot of experiments with approved ethical committee. We can’t perform all experiments again.
For our defense, we listed recent manuscript using 100 uL by intramuscular routes:
https://doi.org/10.1016/j.vaccine.2023.05.071
https://doi.org/10.3389%2Ffimmu.2021.645210
https://doi.org/10.1128%2FCVI.00499-15
https:/doi.org /10.1186/1756-3305-7-145
Based on that and to avoid any problems, we are going to remove the information about volume in the paper. We looked several papers from vaccines without this information.
Reviewer 2: The authors do not answer the question of whether an IFN gamma/IL-10 ratio or IFN gamma/IL-4 ratio is more appropriate as a marker of resistance/susceptibility to L. amazonensis infection
For L.major, for sure IFng/IL-4 is the most indicated, however, for L. amazonensis there is not transcript for Th2 (https://pubmed.ncbi.nlm.nih.gov/30845223), but there are IFNg and IL-10. Based on that, we are more focusing on IFNg/IL-10.
Reviewer 2: In the revised version of the manuscript, the authors continue not to show humoral response results in terms of antibody titer. Therefore, it is not appropriate to mention in the legends in Fig.3 that this is “humoral titer”. They should simply thaw the mouse sera from their experiment and perform a routine titration assay.
Authors: We are very sorry for our mistake. New figure legend was provided.
Reviewer 2: The methods section should be restructured to indicate the conventional subsections where all the tests that have been performed are rigorously explained with the corresponding detail; and not an agglutinating text of several methods as a non-reproducible summary.
We changed as you suggested.
Reviewer 2: If the authors of the study have performed two complementary tests to estimate the parasite load, they should present data from both, including graphs. A correlation cannot be interpreted if the reader is not shown the results of the “limiting dilution test”, including the corresponding figure.
The graphic of LDA was included (Figure 7C).
Reviewer 2: The question has not been answered, the authors describe, again in the revised version (lines 509 and 515), cytokines such as IL-13 that they have not analyzed, which is confusing for the reader.
We were removed.
Reviewer 2: The conclusion section remains incomplete, it does not raise the scientific context (which some of the authors signing this article already published in 2003: doi.org/10.1016/S0264-410X(03)00427-4) of whether or not the formulation of vaccines with killed Leishmania amazonensis parasites is contradictory. Whether it is well standardized. They also do not mention in discussion whether experiments from previous studies in mice and primates have suggested that such formulation (complete antigens of L. amazonensis promastigotes) contains disease-inducing antigens, and their i.m., i.d. or s.c. inoculation resulted in exacerbation of subsequent
Dear reviewer
In my opinion, conclusion is conclusion, must be based just on the results obtained.
About cited manuscript, it is about oral vaccine 2003: doi.org/10.1016/S0264-410X(03)00427-4, that is not work in this manuscript. We already put in the introduction, we included others routes as subcutaneous (ref.:8 ) and intrahepatic enhance (ref.:9).
However, the conclusion is not the place to discuss that.

Round 3
Reviewer 2 Report
Comments and Suggestions for Authors
no comments
Author Response
Dear reviewer,
Thank You for your suggestions.
